# Investigation on the Emission Characteristics with a Wet-Type Exhaust Gas Cleaning System for Marine Diesel Engine Application

**Younghyun Ryu** [1] **, Taewoo Kim** [2] **, Jungsik Kim** [2] **and Jeonggil Nam** [3,*]

1. Division of Marine Mechatronics, Mokpo National Maritime University, Mokpo 58628, Korea; yhryu@mmu.ac.kr
2. Techwin Co., Ltd., Cheongju 28580, Korea; taewoo@techwin.co.kr (T.K.); jskim@techwin.co.kr (J.K.)
3. Division of Marine Engineering, Mokpo National Maritime University, Mokpo 58628, Korea
* Correspondence: jgnam@mmu.ac.kr; Tel.: +82-61-240-7220

**Abstract:** Global air pollution regulations are becoming stricter for large diesel engines powering automobiles and ships. In the automotive sector, Euro 4 regulations came into force in January 2013 in accordance with European Union (EU) emission standards for heavy-duty diesel engines and are based on steady-state testing. In the marine sector, the International Maritime Organization (IMO) Maritime Environment Protection Committee (MEPC) is a group of experts who discuss all problems related to the prevention and control of marine pollution from ships, such as efforts to reduce ozone-depleting substances and greenhouse gas emissions. To reduce the harmful emissions from marine diesel engines, a wet-type exhaust gas cleaning system was developed in this study. As a test, seawater, electrolyzed water, and sodium hydroxide were sequentially injected into the exhaust gas. $SO_2$ was reduced by 98.7–99.6% with seawater injection, NOx by 43.2–48.9% with electrolyzed water injection, and $CO_2$ by 28.0–33.3% with sodium hydroxide injection.

**Keywords:** IMO MEPC; marine diesel engine; emissions; seawater; electrolyzed seawater; sodium hydroxide

## 1. Introduction

Global air pollution regulations for large diesel engines powering automobiles and ships are becoming stricter [1]. In the automotive sector, Euro 4 regulations came into force in January 2013 in accordance with European Union (EU) emission standards for heavy-duty diesel engines based on steady-state testing [1]. EU regulations for exhaust gas emissions have continually become more stringent starting from Euro 1 in 1992 to Euro 4 in 2013 [1]. In the marine sector, Tier 3 regulations enacted by the Marine Environment Protection Committee (MEPC) of the International Maritime Organization (IMO) entered into force on 1 January 2016 [2]. The MEPC is a group of experts who discuss all problems related to the prevention and control of marine pollution from ships, such as efforts to reduce ozone-depleting substances and greenhouse gas (GHGs) emissions [2]. In particular, they review the prevention and regulation of marine pollution by ships and carry out functions related to the adoption and amendment of related international conventions [2]. The IMO headquarters in London, UK, holds the MEPC international conference once or twice a year [2–4]. Many researchers and stakeholders such as marine engine manufacturers, shipbuilding companies, and shipping companies are striving to reduce emissions from marine diesel engines. Marine diesel engines use the lowest-grade fuel and discharge more harmful exhaust gases than other transport diesel engines [5]. Therefore, studies have been conducted to improve the combustion process by modifying the fuel to reduce emissions from

marine diesel engines. Ryu et al. [6–8] attempted to improve the performance and lower the emissions of a diesel engine by mixing dimethyl ether with heavy fuel oil (HFO). Ryu et al. [9–11] also applied two types of fuel additives based on Ca and Fe to reduce the fuel consumption and emissions of marine diesel engines. Geng et al. [12] applied waste cooking oil to a marine diesel engine. Katayama et al. [13] applied Jatropha oil, which is a biofuel, to marine diesel engines. Hashimoto et al. [14] mixed palm oil, which is a biofuel, with gas oil and MDO (marine diesel oil) and investigated the combustion characteristics. Nam et al. [15] applied an emulsion fuel, which is a mixture of fuel and water, to a diesel engine and examined the combustion and emission characteristics. Panomsuwan et al. [16] reduced pollutant emissions by applying non-thermal plasma technology to a marine diesel engine. Exhaust gas recirculation (EGR) is widely used to reduce the NOx emissions of diesel and commercial vehicles [17,18]. There have also been many studies on using EGR in ships. Selective catalytic reduction (SCR) is being applied to large vehicles such as diesel engine trucks and in onshore power plants to reduce NOx [19]. It has also been installed in many ships. Ryu et al. [20] reported the installation of SCR to reduce NOx emissions from ships to meet the IMO MEPC Tier 3 regulations. Although there are regulations on the particulate matter (PM) emissions of automotive diesel engines, there are no such regulations for marine diesel engines yet. However, it is expected that PM regulations will be enacted in the future. Ntziachristos et al. [21] researched the reduction in PM emissions from marine diesel engines. In addition, there have been many discussions on regulations to reduce SOx emissions from ships. Ships operating in a sulfur environmental control area (SECA) are obligated to use fuel with low sulfur content. The use of 1.0% *m/m* low-sulfur fuel was enforced in 2010, and the regulation was strengthened in 2015 to the use of 0.1% *m/m* low-sulfur fuel. In other areas outside an SECA, the use of 3.5% *m/m* low-sulfur fuel has been enforced since 2012. According to Regulation 14.8 of the International Convention for the Prevention of Marine Pollution from Ships (MARPOL) Annex 6, the use of marine fuel oil with 0.5% *m/m* or less sulfur content was enforced for ships operating anywhere in the world from 1 January 2020 (Regulation 14.1.3 of MARPOL Annex 6) [4].

Many measures are being suggested to reduce SOx emissions, such as scrubbers. Table 1 lists the sulfur content in fuel oil required by the IMO. The IMO also controls improvements in energy efficiency and GHGs emissions through various regulations. Table 2 lists the regulations for $CO_2$ reduction. $CO_2$ is a GHG that is cited as the main cause of global warming. Table 2 lists the $CO_2$ emission factors by the type of fuel used in ships [4].

**Table 1.** IMO sulfur requirements [4].

| Outside ECA (Global Requirement) | Inside ECA |
|---|---|
| 4.5% *m/m* prior to 1 January 2012 | 1.5% *m/m* prior to 1 July 2010 |
| 3.5% *m/m* on and after 1 January 2012 | 1.0% *m/m* on and after 1 July 2010 |
| 0.5% *m/m* on and after 1 January 2020 | 0.1% *m/m* on and after 1 January 2015 |

Emission Control Areas (ECA).

**Table 2.** $CO_2$ emission factors (*g/g* fuel) [4].

| Region | Fuel Type | Year | | |
|---|---|---|---|---|
| | | **2012** | **2030** | **2050** |
| | HFO | 3114 | 3114 | 3114 |
| **Global** | LSFO | 3114 | 3114 | 3114 |
| | MGO | 3206 | 3206 | 3206 |
| | LNG | 2750 | 2750 | 2750 |

Heavy Fuel Oil (HFO), Low Sulphur Fuel Oil (LSFO), Marine Gas Oil (MGO), Liquefied Natural Gas (LNG).

Various technologies are available for reducing NOx, SOx, and $CO_2$. This study applied the wet-type exhaust gas cleaning system to diesel engines. This system is an after-treatment technology that can reduce the three harmful exhaust gas components simultaneously. Ryu et al. [22] suggested reducing NOx by injecting electrolyzed water to exhaust gas. In this study, however, seawater, electrolyzed seawater, and sodium hydroxide were sprayed sequentially onto exhaust gas from a diesel engine to reduce NOx, SOx, and $CO_2$. The NOx emitted from a marine diesel engine contains 90–95% NO, which is insoluble. $NO_2$, which is generated from the oxidation of NO, is known to have 20 times higher solubility in water compared to NO [23,24]. Therefore, NO was oxidized to $NO_2$ through the gas–liquid contact of electrolyzed seawater (electrolyzed water) and exhaust gas to reduce NOx generation. An et al. [25] oxidized NO and absorbed $NO_2$ in exhaust gas by spraying a negative strong acidic solution and positive strong basic solution sequentially. These solutions were obtained through membrane electrolysis, which is easily applicable to ships because of the ready supply of seawater. However, membrane electrolysis of seawater has the disadvantage of needing a strong acid with pH 2–3 and generating a large volume of chlorine gas. To address this disadvantage, Kim et al. [23,24] tried to reduce NOx through non-membrane seawater electrolysis. In Ryu et al. [22], Kim et al. [23,24] and An et al. [25], only NOx reduction was presented, but in this paper, NOx, SOx and CO2 reduction were also presented.

An experimental apparatus was planned, designed, and installed for research on NOx and SOx reduction using real seawater in an indirect injection type diesel engine. Reducing $CO_2$ with sodium hydroxide was also researched. This apparatus for reducing exhaust gas with seawater can be used not only on ships but also in thermal power plants by the sea.

## 2. Experimental Apparatus and Method

### 2.1. Engine Test

In this study, a four-stroke diesel engine of the indirect injection (IDI) type was used. This is a small engine with four cylinders that has a compression ratio of 22:1. Table 3 outlines the engine specifications.

**Table 3.** Test engine specifications.

| Item | Description |
|---|---|
| Engine type | D4BB-G3, four-stroke diesel engine |
| Bore × stroke | 91.1 × 100 mm |
| Combustion type | Indirect injection |
| No. of cylinders | 4 inline |
| Displacement volume | 2607 $cm^3$ |
| MCR output | 45 PS @ 1800 rpm |
| Compression ratio | 22:1 |
| Fuel | Diesel oil |

Maximum Continuous Rating (MCR); Pferde-Starke (PS).

Figure 1 shows the entire experimental apparatus used in this study. To remove NOx, SOx, and $CO_2$ generated from engine combustion, real seawater that had not been electrolyzed, electrolyzed seawater (electrolyzed water), and a sodium hydroxide (NaOH) solution diluted to a certain concentration were sprayed sequentially onto exhaust gas discharged from a diesel engine. In general, nitrogen compounds refer to the combination of NO and $NO_2$ [26]. The nitrogen compounds were measured by using 350-Maritime from TESTO. This model can measure the concentrations of NO, $NO_2$, $SO_2$, and $CO_2$ separately with electrochemical cells. The removal rates in the reactor were calculated as follows [22]:

$$Y_a = \frac{c_{a,i} - c_{a,o}}{c_{a,i}} \times 100\%$$ (1)

where a denotes a chemical species such as NO, NO$_2$, or NOx and Ci and Co denote the concentrations at the inlet and outlet, respectively, of the corresponding chemical species. After the electrolyzed water was sprayed, the inlet and outlet concentrations of NO/NO$_2$/NOx were analyzed. Figure 2 shows a schematic diagram of the experimental apparatus. The pH/OPR/temperature sensors were installed in the pipe for electrolyzed water to examine the effects of operating variables such as the temperature and pH of the electrolyzed water on the NOx removal performance. The VSTAR12 model of Orion was used as the sensor. A magnet-type pump (Yusung, NH-3024PH-CV-5HP) was used to spray the electrolyzed water, and an area-type flowmeter (COREA flow, HGF-1-P) was used to measure the flow of the sprayed electrolyzed water. To produce an oxidant through the electrolysis of seawater, which was one of the purposes of this study, a non-membrane electrolysis tank with a capacity of 1.25 kg Cl$_2$/day, produced by T company, was used, and a three-phase 380 V AC was applied through a rectifier (MK power, MK-50200G). This electrolysis tank was designed for the electrolysis of seawater and circulation of electrolyzed water with a structure that prevents the accumulation of scaling or particulate contaminants.

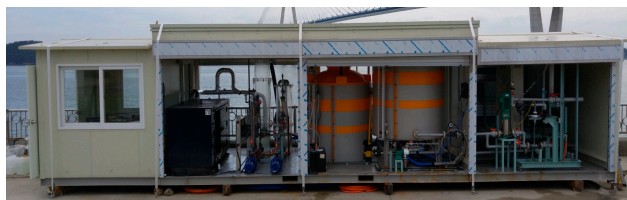

**Figure 1.** Photograph of test facility.

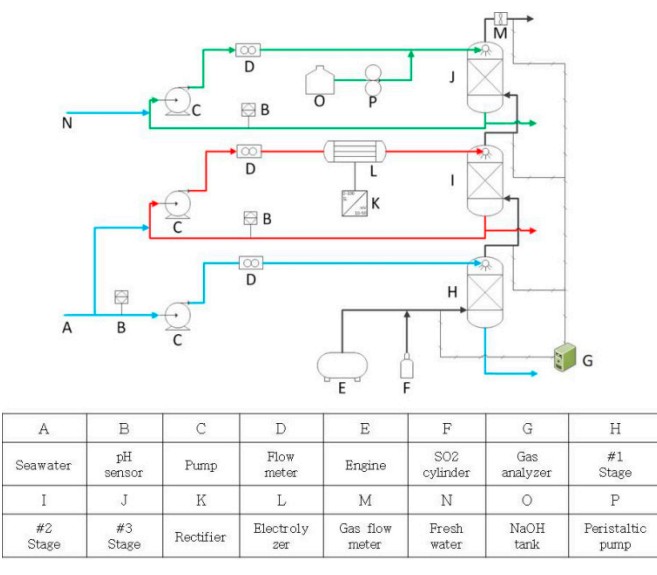

| A | B | C | D | E | F | G | H |
|---|---|---|---|---|---|---|---|
| Seawater | pH sensor | Pump | Flow meter | Engine | SO2 cylinder | Gas analyzer | #1 Stage |
| I | J | K | L | M | N | O | P |
| #2 Stage | #3 Stage | Rectifier | Electrolyzer | Gas flow meter | Fresh water | NaOH tank | Peristaltic pump |

**Figure 2.** Schematic diagram of experimental apparatus.

A scrubber for spraying electrolyzed water was produced as a packed tower with a residence time of 6.5 s, which is given in Equation (2). The filling material for improved gas–liquid contact efficiency was 1-in Tripack made of polypropylene material, and a spiral-type nozzle was used to spray the cleaning solution. Figure 3 shows the cleaning solution spray nozzle and packing. Table 4 lists the test conditions of the experimental apparatus.

$$Residence\ time\ [\text{s}] \ = \ \frac{Volume\ of\ Packing\ \left[\text{m}^3\right]}{Flow\ rate\ of\ gas\ \left[\text{Nm}^3/\text{s}\right]} \tag{2}$$

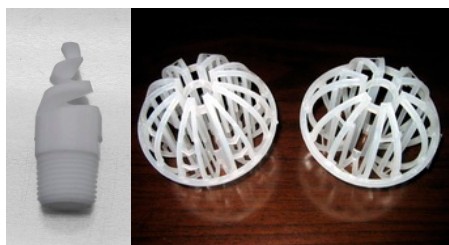

**Figure 3.** Photograph of spray nozzle and packing.

**Table 4.** Test conditions.

| Item | Description |
| --- | --- |
| Spraying order | 1st stage: seawater<br>2nd stage: electrolyzed seawater<br>3rd stage: NaOH (+Na$_2$S) |
| Absorber type | Packed bed |
| Packing | 1″ Tripack (polypropylene) |
| Exhaust gas flow | 130 ± 10 Nm$^3$/h |
| Retention time | 6.3–6.9 s |
| Liquid–gas ratio | 43–50 L/m$^3$ |
| CNO$_{x,i}$ | 887–942 ppmv |
| CSO$_{2,i}$ | 430–470 ppmv |
| CCO$_{2,i}$ | 5.30–5.45 vol.% |
| Cl$_2$ concentration | 4.8–5.3 g Cl$_2$/L |
| NaOH concentration | 6.1–7.7 g NaOH/L |

Table 5 presents the solubility of gases used in this study based on the data provided by Haynes [27].

**Table 5.** Solubility of selected gases in water.

| Gas | Solubility in Water [mol/L] |
| --- | --- |
| NO | $1.51 \times 10^{-6}$ |
| NO$_2$ | $2.44 \times 10^{-5}$ |
| SO$_2$ | $1.25 \times 10^{-3}$ |
| CO$_2$ | $3.48 \times 10^{-5}$ |
| Cl$_2$ | $8.20 \times 10^{-5}$ scientific notion |

## 2.2. Oxidation and Reduction Agent

In this study, the oxidant was produced in real time through the electrolysis of seawater. The quantity of oxidant can be converted to the effective chlorine concentration (g Cl$_2$/L), which is the quantities of chlorine (Cl$_2$), hypochlorite ions, and hypochlorous acid (HOCl) converted to the equivalent amount of Cl$_2$. This was measured with the following titration method:

(1) Inject 1 mL of the sample into 25 mL of ultrapure water.
(2) Add 1 g of potassium iodide and 1 mL of acetic acid.
(3) Add two to three drops of 1% starch solution. The solution then turns reddish brown.
(4) Add 0.1 N sodium thiosulfate to the reddish-brown solution and measure the amount added until it becomes colorless.
(5) Determine the effective chlorine concentration as follows:

$$\text{Available chlorine}[\text{g } Cl_2/\text{L}] = \frac{C[\text{mL}] \times 3.545}{S[\text{mL}]} \tag{3}$$

where $C$ is the titration amount of 0.1 N sodium thiosulfate and $S$ is the sample injection amount.

In the No. 3 tower, sodium hydroxide diluted in water was sprayed as absorbent, and the concentration was measured according to the following procedure:

(1) Add 1 mL of sample to 25 mL of ultrapure water.
(2) Add two to three drops of phenolphthalein. The solution turns red.
(3) Add 0.1 M hydrochloric acid to the red solution and measure the amount injected until it becomes colorless.
(4) Determine the concentration of sodium hydroxide as follows:

$$\text{Sodium hydroxide}[\text{g NaOH/L}] = \frac{C[\text{mL}] \times 4}{S[\text{mL}]} \tag{4}$$

where *C* is the titration amount of 0.1 M hydrochloric acid and *S* is the sample injection amount.

Figure 4 shows the calculated concentrations of the oxidant and absorbent according to the operation time.

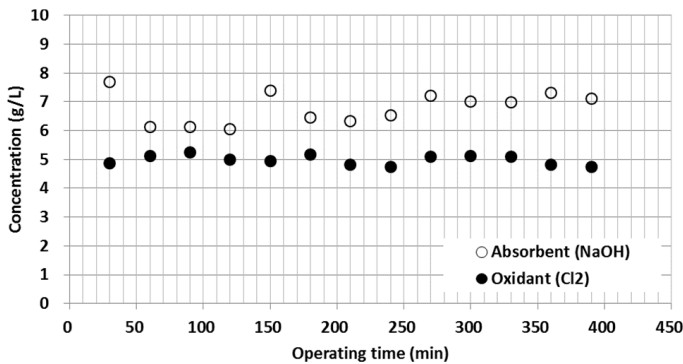

**Figure 4.** Concentrations of oxidant and absorbent against the operating time.

## 3. Results and Discussion

### 3.1. Sulfur Oxide (SOx) Emissions

$SO_2$ gas was removed by using the alkalinity of seawater, as suggested by Andreasen et al. [28]. After $SO_2$ dissolves in water, it exists as bisulfate ions ($HSO_3^-$) and sulfate ions ($SO_3^2$). As a result, the solution reaches equilibrium because of the decreased pH, and the dissolved $CO_2$ is discharged. In this study, the diluted solutions of seawater, electrolyzed water, and sodium hydroxide were sequentially sprayed in each stage. The main purpose of the No. 1 tower, in which seawater was sprayed, was to remove $SO_2$ and particulate contaminants. When a large amount of particulate contaminants in the exhaust gas flows into the No. 2 tower, which operates in a closed loop, this can negatively influence the efficiency of the electrolysis tank. When $SO_2$ is absorbed, it decreases the pH of the cleaning water. When the pH of the electrolyzed water decreases to neutral or lower, the dissolved $Cl_2$ can be discharged. This implies the loss of oxidant by energy input. Therefore, prior removal of and particulate contaminants in the No. 1 tower can minimize the above adverse effects. Figure 5 illustrates the $SO_2$ removal rate of this system. The $SO_2$ removal rate was 98.7–99.6%, which indicates relatively stable operation. This confirms that at least 98% of $SO_2$, which has high solubility in water, was removed with the No. 1 tower, as shown in Figure 6.

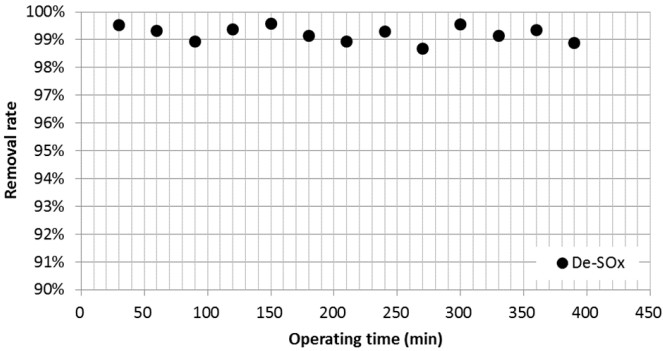

**Figure 5.** Removal rate of $SO_2$.

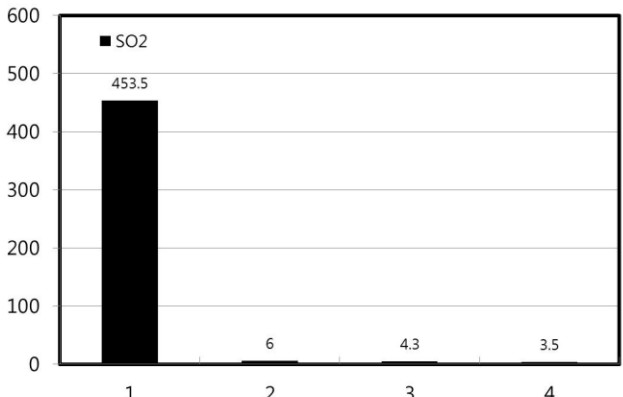

**Figure 6.** Average concentration of $SO_2$ at each stage.

$SO_2$ removal mechanism:

$$SO_{2(g)} \Leftrightarrow SO_{2(aq)} \tag{5}$$

$$SO_{2(aq)} + H_2O_{(l)} \Leftrightarrow HSO_{3(aq)}^- + H_{(aq)}^+ \tag{6}$$

$$HSO_{3(aq)}^- + H_2O_{(l)} \Leftrightarrow SO_{3(aq)}^{2-} + H_{(aq)}^+ \tag{7}$$

$$HCO_{3(aq)}^- + H_{(aq)}^+ \Leftrightarrow CO_{2(aq)} + H_2O_{(l)} \tag{8}$$

$$CO_{2(aq)} \Leftrightarrow CO_{2(g)} \tag{9}$$

### 3.2. Nitric Oxide (NOx) Emissions

Wet-type NOx removal involves the oxidation of NO, which accounts for 90–95 vol% of NOx. $Cl_2$ was used as the oxidant of NO in this study; it was produced by non-membrane electrolysis of NaCl in water. The $Cl_2$ gas produced from the electrode surface quickly dissolves and exists as three chemical species. As shown in Figure 7, each chemical species reaches equilibrium according to the pH. When the pH is lower, the condition becomes more unstable, and gases can be discharged. HOCl, which is predominant in the weak acid domain, has an oxidizing power at least 80 times higher than that of $OCl^-$ [29]. In this study, therefore, the condition of pH 6–7 was maintained so that HOCl would not be discharged as gas while maintaining the oxidizing power of the electrolyzed water, which was the cleaning solution for the No. 2 tower. The reducing agent sodium sulfide ($Na_2S$) was injected in the No. 3 tower to improve the NOx removal performance, and its effect was observed.

The results indicated that the total NOx removal rate was 43.2–48.9%, as shown in Figure 8, and the injection of the reducing agent in the No. 3 tower showed a negative effect. To express the NO and $NO_2$ removal rates separately, as shown in Figure 9, the NO removal rate decreased after injection of the reducing agent. This can be interpreted that the strong reducing agent $Na_2S$ reduced $NO_2$ or $NO_3^-$ to NO. Figure 10 shows the average concentrations of NO and $NO_2$ at the outlet of each stage.

The oxidation reaction of NO was dominant in the No. 2 tower, whereas the absorption of $NO_2$ was dominant in the No. 3 tower. In the case of the No. 3 tower, in which the NaOH solution was sprayed, the NO oxidation rate was around 17%, unlike the No. 1 tower. This appears to be because both NO and $NO_2$ were removed by reaction with NaOH.

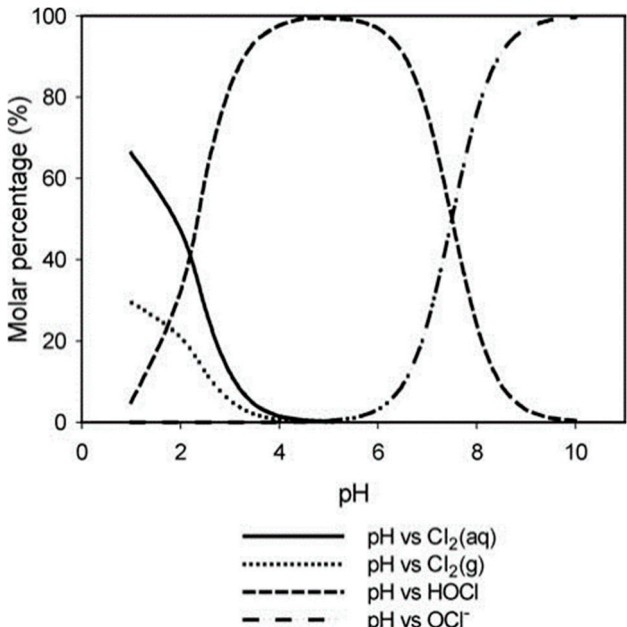

**Figure 7.** Chlorine speciation profile as a function of pH.

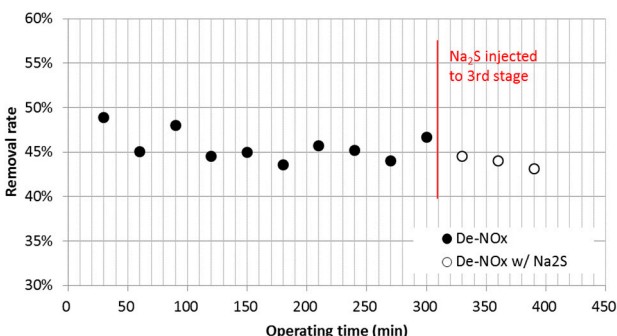

**Figure 8.** Removal rate of $NO_x$.

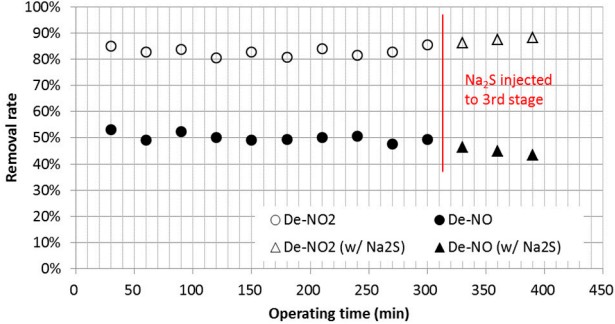

**Figure 9.** Removal rate of $NO/NO_2$.

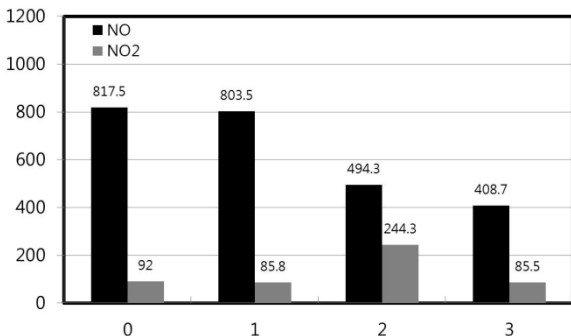

**Figure 10.** Average concentration of NO/NO$_2$ at each stage (without Na$_2$S).

NO/NO$_2$ removal mechanism by electrolyzed seawater [25,30–33]:

$$NO_{(g)} \Leftrightarrow NO_{(aq)} \tag{10}$$

$$NO_{(aq)} + OCl^-_{(aq)} \rightarrow NO_{2(aq)} + Cl^-_{(aq)} \tag{11}$$

$$4NO_{2(aq)} + 2H_2O_{(l)} + O_{2(g)} \rightarrow 4HNO_{3(aq)} \tag{12}$$

$$2NO_{(g)} + O_{2(g)} \rightarrow N_2O_{4(g)} \tag{13}$$

$$N_2O_{4(g)} \Leftrightarrow N_2O_{4(aq)} \tag{14}$$

$$N_2O_{4(aq)} + H_2O_{(l)} \rightarrow HNO_{3(aq)} + HNO_{3(aq)} \tag{15}$$

NO/NO$_2$ removal mechanism by NaOH [25,30–33]:

$$NO_{(aq)} + NO_{2(aq)} + 2NaOH_{(l)} \rightarrow NaNO_{2(aq)} + NaNO_{3(aq)} + H_2O_{(l)} \tag{16}$$

$$NO_{(aq)} + NO_{2(aq)} + 2NaOH_{(l)} \Leftrightarrow 2NaNO_{2(aq)} + H_2O_{(l)} \tag{17}$$

$$2NO_{2(aq)} + 2NaOH_{(l)} \Leftrightarrow NaNO_{2(aq)} + NaNO_{3(aq)} + H_2O_{(l)} \tag{18}$$

NO/NO$_2$ removal mechanism by Na$_2$S [34–36]:

$$S^{2-}_{(aq)} + H_2O_{(l)} \Leftrightarrow HS^-_{(aq)} + OH^- \tag{19}$$

$$NO_{2(aq)} + HS^-_{(aq)} \rightarrow NO^-_{2(aq)} + HS \tag{20}$$

$$2NO_{2(aq)} + Na_2S_{(s)} \rightarrow N_{2(g)} + Na_2SO_{4(aq)} \tag{21}$$

$$3Na_2S_{(s)} + 8HNO_{3(aq)} \rightarrow 6NaNO_{3(aq)} + 3S_{(s)} + 2NO_{(g)} + H_2O_l \tag{22}$$

### 3.3. Carbon Dioxide (CO$_2$) Emissions

CO$_2$ dissociates into bicarbonate ions (HCO$_3{}^-$) and carbonate ions (CO$_3{}^{2-}$) after being dissolved in water in the carbonate system, and they reach equilibrium according to the concentration of hydrogen ions. The hydrogen ion concentration of the cleaning solution was increased by the sodium hydroxide used to remove NO and NO$_2$. As a result, the equilibrium of the carbonate system moved in the direction of dissolving, and CO$_2$ was converted to CO$_3{}^{2-}$ [37,38]. Consequently, a CO$_2$ removal rate of 28.0–33.3% was achieved, as shown in Figure 11. As shown by the average CO$_2$ concentration at each stage in Figure 12, the CO$_2$ concentration did not change in the No. 1 and No. 2 towers but decreased in the No. 3 tower. O$_2$ absorption did not occur in the seawater spray tower and electrolyzed seawater spray tower because HCO$_3{}^-$ and CO$_3{}^{2-}$ exist in general seawater from the hydrogen ions reaching equilibrium with CO$_2$ in the atmosphere.

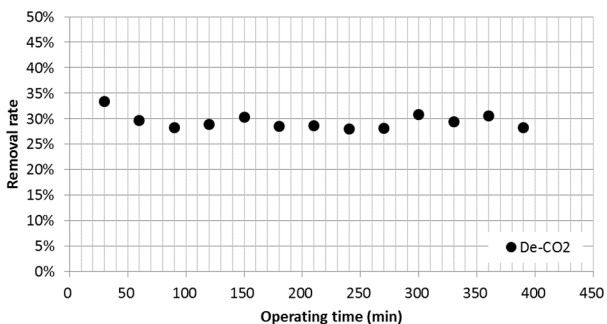

**Figure 11.** Removal rate of $CO_2$.

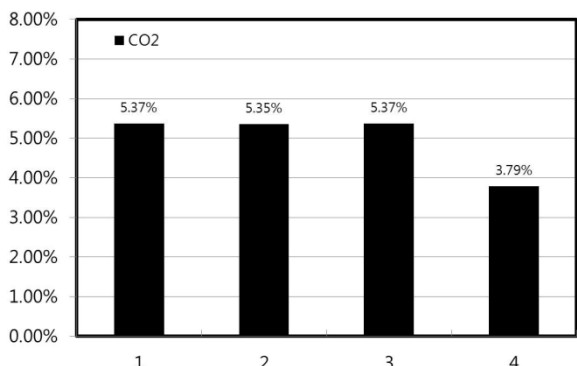

**Figure 12.** Average concentration of $CO_2$ at each stage.

$CO_2$ removal mechanism:

$$CO_{2(g)} \Leftrightarrow CO_{2(aq)} \tag{23}$$

$$CO_{2(aq)} \ + \ H_2O_{(l)} \Leftrightarrow HCO^-_{3(aq)} \ + \ H^+_{(aq)} \tag{24}$$

$$CO_{2(aq)} \ + \ OH^-_{(aq)} \Leftrightarrow HCO^-_{3(aq)} \tag{25}$$

$$CO^{2-}_{3(aq)} \ + \ H^+_{(aq)} \Leftrightarrow HCO^-_{3(aq)} \tag{26}$$

$$HCO^-_{3(aq)} \ + \ OH^-_{(aq)} \Leftrightarrow CO^{2-}_{3(aq)} \ + \ H_2O_{(l)} \tag{27}$$

## 4. Conclusions

In this study, a wet-type exhaust gas cleaning system, which is an after-treatment device for reducing exhaust gases from a marine diesel engine, was developed and evaluated. An experiment was performed in which seawater, electrolyzed seawater, and sodium hydroxide were sequentially sprayed onto the exhaust gas. The following results were obtained:

(**1**) Electrolyzed water was sprayed into the second tower with NOx as the oxidizer, and the oxidizer concentration stayed within the range of 4.8–5.3 g $Cl_2$/L during the operating period. Sodium hydroxide was injected into the third tower with NOx as the absorbent, and the concentration of the absorbent during the operation stayed within the range of 6.1–7.7 g/L.

(**2**) $SO_2$ in the exhaust gas from the diesel engine was decreased by 98.7–99.6%. Among the three towers, the first tower spraying seawater was confirmed to remove 98.7% of the $SO_2$.

(**3**) NOx decreased by 43.2–48.9%. The oxidation rate of NO was higher in the second tower spraying electrolyzed water, and $NO_2$ was absorbed in the third tower spraying sodium hydroxide. Sodium hydroxide was found to be involved not only in $NO_2$ absorption but also in direct absorption of NO. Na2S, used to improve the NOx removal rate, was found to have a negative effect.

(**4**) $CO_2$ was reduced by 28.0–33.3%, and the third tower spraying sodium hydroxide was confirmed to have an average removal rate of 29.3%. In the future, additional research on scrubbing technology using seawater will be conducted to increase the NOx reduction rate.

**Author Contributions:** Conceptualization, Y.R.; Methodology, Y.R., T.K., J.K. and J.N.; Validation, Y.R.; Formal Analysis, T.K., J.K. and J.N.; Investigation, Y.R., T.K. and J.K.; Writing—Original Draft Preparation, Y.R.; Writing—Review and Editing, T.K., J.K. and J.N.; Visualization, Y.R.; Supervision, J.N.; Project Administration, J.N. All authors have read and agreed to the published version of the manuscript.

**Funding:** This study did not receive external funding.

**Conflicts of Interest:** The authors declare that there is no conflict of interest.

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
