# Peer review of "Investigation on the Emission Characteristics with a Wet-Type Exhaust Gas Cleaning System for Marine Diesel Engine Application"

_jmse, doi:10.3390/jmse8110850_

Round 1
Reviewer 1 Report
The article deals with Emission Characteristics with a Wet-Type Exhaust Gas Cleaning System for Marine Diesel Engine Application. The article needs extensive revision before it can be considered for publication. My concerns are as follows: 1. The abstract contains too much background information. Condense that. 2. Add more information on what tests were performed and how they were performed in the abstract. 3. Add more results in the abstract. 4. What is the motivation of current research? This is not explicitly mentioned. 5. Discuss those papers referred here individually. For example: "Ryu et al. [6-8] attempted to improve the performance and lower the emissions of a diesel engine by mixing dimethyl ether with heavy fuel oil (HFO). Ryu et al. [9-11] also applied two types of fuel additives based on Ca and Fe to reduce the fuel consumption and emissions of marine diesel engines." "To address this disadvantage, Kim et al. [23, 24] tried to reduce NOx through non-membrane seawater electrolysis". Do not lump references here. 6. Add results of the papers discussed in the introduction section. 7. Do not present result in the introduction: "In this study, real seawater was sprayed onto actual exhaust gas discharged from a diesel engine, and an approximately 99% reduction of SOx was achieved. Furthermore, an approximately 30% reduction of CO2 was achieved by spraying sodium hydroxide onto exhaust gas. " 8. Section 3.1 should go to materials and method. 9. Statistical analysis of data must be performed in a scientific article. 10. Authors should compare the results with the results in the existing literature, not just describing their result. This is a scientific article.Author Response
"Please see the attachment."

Reviewer 2 Report
1. Please indicate the uncertainty of the measuring instruments in this study.
2. Please comment on the possibility of using the proposed methods for engines of other dimensions.
3. Please analyze the feasibility of meeting the promising environmental regulations by using the proposed methods.
4. Please indicate the further development of this line of research in the "Conclusion" section.
Author Response
"Please see the attachment."

Round 2
Reviewer 1 Report
Please address the comments below:
- "As a test, seawater, electrolyzed water, and sodium hydroxide were sequentially injected into the exhaust gas." At what conditions the marine diesel engine was operated?
- Point 7: Do not present result in the introduction: "In this study, real seawater was sprayed onto actual exhaust gas discharged from a diesel engine, and an approximately 99% reduction of SOx was achieved. Furthermore, an approximately 30% reduction of CO2 was achieved
by spraying sodium hydroxide onto exhaust gas. "
Response 7: The authors believe it is necessary to present some findings in the introduction section. I have reviewed hundreds of papers in my entire career and never have I encountered such arrogance. Please go through the articles in high-quality journals. - Point 10: Authors should compare the results with the results in the existing literature, not just describing their result. This is a scientific article.
Response 10: Thanks for the great comment. As the authors mentioned in comment #5, the authors think that pre-treatment technology and post-treatment technology should be used at the same time in order to reduce exhaust gas from ships. Thank you for your understanding. Have you actually read through my comments and intended to address any time during the revision process?
Author Response
"Please see the attachment."

Reviewer 2 Report
The article can be accepted for publication.
Author Response
"Please see the attachment."

Round 3
Reviewer 1 Report
The authors must remove all self-citations.